The relationship between mitochondrial DNA haplotype and its copy number on body weight and morphological traits of Wuliangshan black-bone chickens

Li Wenpeng 1
Yang Zhen 1
Yan Chao 2
Chen Siyu 3 chensiyu@fosu.edu.cn
Zhao Xingbo 1 zhxbcau@126.com
1 College of Animal Science and Technology, China Agricultural University , Beijing , China
2 Agricultural Genomics Institute, Chinese Academy of Agricultural Science , Shenzhen , China
3 School of Life Science and Engineering, Foshan University , Guangdong , China
Li Zhiming
Electronic publication date: 2024 Dec 16
Publication date: 2024
Volume: 12
Electronic Location ID: e17989
Received 2023 Dec 6; Accepted 2024 Aug 7
Copyright: © 2024 Li et al.
Copyright year: 2024
Copyright holder: Li et al.
License: This is an open access article distributed under the terms of the Creative Commons Attribution License, which permits unrestricted use, distribution, reproduction and adaptation in any medium and for any purpose provided that it is properly attributed. For attribution, the original author(s), title, publication source (PeerJ) and either DOI or URL of the article must be cited.
License URL: https://creativecommons.org/licenses/by/4.0/

Keywords: Wuliangshan black-bone chicken, Mitochondrial DNA, Haplotype, Phenotype, D-loop

Funding: National Natural Science Foundation 32102596 This work was funded by the National Natural Science Foundation (32102596). The funders had no role in study design, data collection and analysis, decision to publish, or preparation of the manuscript.

==============================
Mitochondria play a pivotal role as carriers of genetic information through their circular DNA molecules. The rapid evolution of the D-loop region in mitochondria makes it an ideal molecular marker for exploring genetic differentiation among individuals within species and populations with close kinship. However, the influence of mtDNA D-loop region haplotypes and mtDNA copy numbers on phenotypic traits, particularly production traits in chickens, remains poorly understood. In this comprehensive study, we conducted D-loop region amplification and sequencing in the blood mitochondria of 232 female Wuliangshan black-bone chickens. Our investigation identified a total of 38 haplotypes, with a focus on 10 haplotypes that included more than five individuals. We meticulously analyzed the correlations between these haplotypes and a range of traits, encompassing body weight, tibial length, tibial circumference, body oblique length, chest width, and chest depth. The results unveiled significant disparities in specific tested traits across different haplotypes, indicating a tangible association between mtDNA haplotypes and traits in chickens. These findings underscore the potential impact of mitochondrial DNA variations on energy metabolism, ultimately leading to divergent chicken phenotypes. Furthermore, our examination revealed positive correlations between mtDNA copy numbers and tested traits for select haplotypes, while other haplotypes exhibited non-uniform relationships between traits and mtDNA copy numbers. In addition, phylogenetic analysis disclosed the involvement of two subspecies of red jungle chicken in the origin of Wuliangshan black-bone chickens. Consequently, our research contributes novel insights into mitochondrial genomic selection, augments comprehension of the roles played by haplotypes and mtDNA copy numbers in chicken population genetics and phylogenetic analysis, and furnishes fundamental data crucial for the preservation and provenance determination of black-bone chickens.

Introduction

Mitochondria are indispensable cytoplasmic organelles within eukaryotic cells, playing a crucial role in maintaining cellular energy homeostasis and metabolism through the production of adenosine triphosphate (ATP) (Francis & Gavin, 2013). Additionally, mitochondria are vital carriers of genetic information, containing circular DNA molecules. In higher animal species, mitochondrial DNA (mtDNA) is a covalently closed double-stranded molecule, approximately 16.5 kb in length, and is inherited exclusively through the maternal lineage (Birky, 1995). Chicken mtDNA, similar to that of mammals, encodes 37 gene products, including 13 oxidative phosphorylation subunits (complexes I, III, IV, and V) (Di Lorenzo et al., 2015).

Despite its relatively simple gene organization, mtDNA evolves at a rate approximately 5 to 10 times faster than single-copy nuclear DNA, primarily driven by mutations rather than recombination. Notably, the control region, known as the D-loop region, is non-coding but exhibits a faster evolution rate compared to the coding regions. Consequently, the D-loop region has emerged as an ideal molecular marker for studying genetic differentiation among individuals within species and populations, particularly those of close kinship (Jia et al., 2022). Recent studies have increasingly focused on the interplay between mtDNA haplotypes and phenotypes in farm animals. For instance, mtDNA haplotypes, determined based on D-loop region sequences, have shown potential for influencing traits such as meat quality in cattle (Kim et al., 2009; Mannen et al., 2003), milk quality and yield in dairy cattle (Tsai & John, 2016), and ewe prolificacy in sheep (Reicher et al., 2012). In chickens, mtDNA haplotypes have also been investigated in relation to traits such as pectoral muscle fat content, duodenum length, and body weight (Lu et al., 2016). Furthermore, specific mtDNA haplotypes have been associated with disease susceptibility, significantly impacting outcomes in infections such as Marek’s disease virus (MDV) (Silva et al., 2019; Wakenell et al., 1996). Nonetheless, whether mtDNA haplotypes in chickens contribute to the enhancement of production traits and overall health remains an open question.

The mtDNA copy number represents the quantity of mitochondrial genomes within a cell, with each mitochondrion containing multiple copies of the mitochondrial genome, which often varies by tissue type. For instance, in mice, the heart exhibits a higher mtDNA copy number compared to the spleen, reflecting differences in energy metabolism intensity across various tissues (Masuyama et al., 2005). Notably, alterations in mitochondrial energy metabolism are closely associated with mtDNA copy number, as increased mtDNA copy number can compensate for impaired mitochondrial respiratory function (Kelly et al., 2012). For example, prenatal betaine administration has been shown to increase mtDNA copy number, protecting chickens from corticosterone-induced fatty liver (Yun et al., 2017). Moreover, altered mtDNA copy number have been linked to disease susceptibility, such as the mtDNA ND6 P25L mutation, which impairs retinal function in mice (Lin et al., 2012). Additionally, reduced mtDNA copy number exacerbate mitochondrial abnormalities in spermatocytes and spermatids in the testes, whereas increased copy number normalize testis morphology and proteomic changes in spermatocytes (Jiang et al., 2017).

To date, research on mtDNA copy number has predominantly focused on their role in regulating various diseases, including mitochondrial disorders, neurodegenerative conditions, cancer, and aging in humans. Our previous study reported that specific mtDNA haplotypes induce functional differences in cellular energy metabolism in Gallus gallus domesticus (Kong et al., 2020), though it did not further classify their effects at the individual production level. Black-bone chickens, renowned for their distinctive black skin, black bones, and black muscles, boast a rich historical legacy. Unlike their domestic counterparts, black-bone chickens possess unique medicinal properties and were first documented in the Compendium of Materia Medica, an ancient Chinese herbal medicine volume published in 1578 AD. Previous studies have explored genetic diversity, phylogenetic relationships, and origins in some Chinese black-bone chickens and Japanese chickens, often employing mitochondrial D-loop regions or microsatellite markers (Rowshan et al., 2011; Zhou et al., 2010). Generally, studying mitochondrial DNA’s influence necessitates ensuring that subjects belong to the same breed or share a similar nuclear genome. Interestingly, black-bone chickens exhibit a high level of genetic diversity (Huang et al., 2021). Wuliangshan black-bone chicken represents one of the native breeds. However, the conservation and selective breeding of the Wuliangshan black-bone chicken breed have lagged, with a deficiency in high-quality technological innovation. This situation has arisen due to a long-term lack of selective breeding and matching, compounded by issues such as inbreeding and the introduction of genes from foreign breeds. These factors have led to difficulties in maintaining the purity of the breed’s genes, and its genetic lineage has become less clear. Consequently, investigating its origins is of significant importance to safeguard its germplasm resources.

Given these considerations, we hypothesized that mtDNA copy number significantly influences phenotype development. Consequently, we aimed to elucidate the effects of mtDNA haplotypes and their copy number on various traits, including body weight, tibial length, tibial circumference, body oblique length, chest width, and chest depth in Wuliangshan black-bone chickens. Previous research suggests that Chinese game chickens originated from multiple subspecies of red junglefowl and underwent several independent domestication events. Moreover, black-bone chickens are believed to primarily descend from three subspecies of red junglefowl (Zhu et al., 2014). In this study, we investigated the origin and systematic evolution of Wuliangshan black-bone chickens by comparing additional mtDNA D-loop partial sequences with those from 13 Chinese native chicken breeds and two subspecies of red junglefowl within the local area. Our findings aim to provide novel insights into mitochondrial genomic selection and enhance our understanding of the roles of haplotypes and mtDNA copy number in population genetics and phylogenetic analyses in chickens. Ultimately, this research will contribute to the conservation of Wuliangshan black-bone chickens and clarify their origins.

Materials and Methods

Ethical statement

The experimental protocols and animal care were formally approved by the China Agricultural University Laboratory Animal Welfare and Animal Experimental Ethical Inspection (approval number: CAU20180619-5).

Sample collection and DNA extraction

A total of 232 blood samples were randomly collected from a population of 2,000 Wuliangshan black-bone chickens, a native Chinese breed known for its dual-purpose production characteristics. These samples were procured from a farm situated in Dali, Yunnan Province, China. All sampled birds were exclusively females and were provided unrestricted access to diets and water, while being raised under natural light cycles and ambient temperatures. They were collectively housed in 20-square-meter enclosures at a density of two birds per square meter. The dietary regimen consisted of commercial concentrates, with feed and water available ad libitum. Body measurements, including tibial length, tibial circumference, body oblique length, chest width, and chest depth, were meticulously recorded at the age of 42 weeks. To facilitate subsequent procedures, feed and water were withheld for a period of 12 h prior to sample collection. Body weight measurements were conducted using precision electronic balances (in grams), while other morphometric traits were assessed with digital calipers during the slaughter process. To promote animal welfare, all slaughter procedures were carried out after electrical stunning by staff with at least two years of work experience who were blind to the aim of this study. These measurements adhered to established methods (Yang et al., 2022). Approximately 2 mL of blood was collected using BD 6 mL EDTA K2 anticoagulation vacuum blood collection tubes. The collected blood samples were promptly placed on dry ice and subsequently stored at −80 °C until further analysis. Genomic DNA was extracted using a Tissue/Cell Genome DNA Rapid Extraction Kit (Aidlab, Beijing, China) following the manufacturer’s instructions. The quality (260/280 ratio) and quantity of the genomic DNA were assessed using a TECAN Infinite200 PRO (TECAN, Männedorf, Switzerland). Sequencing of the D-loop region was performed to establish haplotypes.

PCR amplification and DNA sequencing

For the amplification of the D-loop region, the following primer sequences were employed: L16750: 5′-AGGACTACGGCTTGAAAAGC-3′; H547: 5′- ATGTGCCTGACCGAGGAAC-CAG-3′ (Liu et al., 2006). Real-time quantitative PCR (RT-qPCR) was carried out using 1 µL of total DNA in a 20 µL reaction mixture comprising 10 µL of 2X TaqMan universal PCR master mix, 1 µL of primer, and nuclease-free water. The reactions were conducted in a 96-well plate using the Bio-Rad CFX-96 system (Bio-Rad, Hercules, CA, USA), with thermal cycling conditions consisting of 5 min at 95 °C, followed by 35 cycles of 20 s at 95 °C, 30 s at 59 °C, and 30 s at 72 °C. The sequencing of PCR products was carried out by Sangon Biotech (Beijing, China).

Relative mtDNA copies detection

The quantification of relative mtDNA copy number was achieved using real-time PCR. Primer sequences for ND2 were as follows: F: 5′-CCTAATCGGAGGCTGAATG-3′; R: 5′-GGTGAGAATAGTGAGTTGTGGG-3′. The single-copy gene VIM was selected as the reference for standardization, which is considered a stable and ubiquitous gene across different cell types and conditions. Primer sequences for VIM were as follows: F: 5′-CAGCCACAGAGTAGGGTAGTC-3′; R: 5′-GAATAGGGAAGAACAGGAAAT-3′ (Juan et al., 2013). RT-qPCR was conducted using 1 µL of total DNA in a 20 µL reaction mixture comprising 10 µL of 2X TaqMan universal PCR master mix, 1 µL of primer, and nuclease-free water. The reactions were executed in a 96-well plate using the Bio-Rad CFX-96 system (Bio-Rad, Hercules, CA, USA), with the following thermal cycling conditions: 5 min at 95 °C, followed by 35 cycles of 20 s at 95 °C, 30 s at 59 °C, and 30 s at 72 °C. The amplification efficiency approached 100%, with a minimal difference of less than 5% in amplification efficiency between the two primers. Triplicate amplifications were performed for all samples. The mtDNA copy number of each sample was compared by calculating the ratio of mitochondrial to nuclear DNA abundance (mtDNA/nDNA) (Wang et al., 2017). Three technical replicates per sample were used.

Haplotype analysis

Following multiple sequence alignment of the D-loop region sequences using MEGA-X software (Kumar et al., 2018), variant site sequences were extracted, haplotypes were consolidated, and sequence polymorphism was analyzed utilizing DnaSPv6.12.3 software (Rozas et al., 2017). Phenotypic differences among haplotypes with a sample size exceeding five individuals were rigorously assessed. To investigate the origin and systematic evolution of Wuliangshan black-bone chickens, additional mtDNA D-loop partial sequences from two subspecies of red junglefowl and 13 Chinese native chicken species were retrieved from GenBank (Table 1). An NJ (neighbor-joining method) molecular phylogenetic tree, with a bootstrap value set at 1,000, was meticulously constructed employing MEGA-X. This comprehensive analysis encompassed 49 haplotypes of mtDNA D-loop regions, incorporating two subspecies of red junglefowl as an outgroup, 13 Chinese native chicken breeds, and the Wuliangshan black-bone chickens sampled in this study.

Table 1 D-Loop region sequence of different breeds obtained from GenBank.

Breed	Accession no. in GenBank	Author	Collection site	
Chahua (CH1-3)	AF512085, AF512089	Y. P. Liu, et al.	Yunnan, China	
MK163564	Kong, M. and Zhao, X.	
Yanjing black-bone (YJ1-3)	AF512324, AF512326, AF512327	Y. P. Liu, et al.	Yunnan, China	
Qinyuan (QY)	AF512260	Y. P. Liu, et al.	Guangdong, China	
Gushi black-bone (GS1-5)	AF512144, AF512145, AF512146, AF512150	Y. P. Liu, et al.	Henan, China	
GU261678	Miao, Y. W, et al.	
Baiyiner (BYE)	AF128322	Y. P. Liu, et al.	Jiangxi, China	
G. g. spediceus (GGS1-5)	AF512182, AF512185, AF512186, AF512187, AF512188	Y. P. Liu, et al.	Myanmar	
G. g. gallus (GGG1-13)	AB007720, AB007725, AB007756	Miyake, T.	Unknown	
AB009440, AB009439	Sumatra	
AB009438, AB009437	Lombok	
AB009435, AB009434	Vietnam	
AB009433	Philippines	
AB009432	Thailand	
AB007752	Unknown	
AB007757		
Dagu (DG)	KT283576	Gu, J. and Li, S.	Liaoning, China	
Emei black (EMB)	MT555047	Gu, J.	Sichuan, China	
Hetian (HT)	MT555048	Gu, J.	Fujian, China	
Shouguang (SG)	MK163561, MK163560	Kong, M. and Zhao, X.	Shandong, China	
Taihe black-bone (TH)	AB086102	Wada, Y.	Jiangxi, China	
Taoyuan (TY)	KF981434	L. Liu, et al.	Hunan, China	
Wuding (WD)	GU261676	Y. W. Miao, et al.	Yunnan, China	
Xianju (XJ)	GU261677	Y. W. Miao, et al.	Zhejiang, China	

Data analysis

The statistical analysis of the association between haplotypes and phenotypes was meticulously conducted employing the statistical variance method. The presentation of results adhered to the format of mean ± standard error of the mean (SEM), and one-way analysis of variance (ANOVA) was conscientiously applied utilizing SPSS v26.0 software (IBM, SPSS v 26.0, Armonk, NY, USA). PEARSON correlation coefficients were computed to assess the correlations between mtDNA copies and haplotypes with the tested traits. Multiple comparisons were judiciously addressed using Fisher’s least significant difference (LSD) method. A significance threshold of P < 0.05 was rigorously adhered to for all statistical assessments.

Results

Correlations between haplotypes and traits

The sequences (504 bp/animal) obtained were at the position from 4 to 508 relative to the reference sequence (AB007757). All the mitochondrial non-synonymous mutations categorized the Wuliangshan black-bone chickens into a total of 38 haplotypes. We conducted a comprehensive phenotypic analysis on 10 haplotypes, each encompassing more than five individuals, and were identified with 27 polymorphic sites (Table 2). Our findings unveiled significant distinctions among these haplotypes concerning various production traits.

Table 2 Haplotype list showing nucleotide substitutions observed in mitochondrial DNA D-Loop in Wuliangshan black-bone chicken.

		Position of nucleotide substitution relative to GenBank GenBank: AB007757.1 as reference sequence	
H	N	14	19	167	212	217	221	225	234	236	239	242	243	246	254	256	261	281	296	302	310	315	317	322	342	363	367	446	
R		A	A	T	A	T	C	C	C	T	A	G	T	T	T	T	C	A	C	C	C	T	A	T	A	C	T	C	
3	37	–	A	C	G	T	C	T	C	T	A	G	T	C	T	T	C	A	C	C	C	C	A	T	A	C	T	C	
4	35	–	–	T	A	T	C	C	C	T	A	G	T	T	T	T	C	A	T	C	C	T	A	T	A	C	T	C	
5	6	–	A	T	A	T	T	C	C	T	A	G	C	C	T	C	T	A	A	T	T	T	A	C	A	C	T	T	
6	10	–	A	T	G	C	C	C	C	T	A	G	C	C	T	C	T	A	C	C	T	C	A	T	A	C	T	T	
7	28	–	A	T	G	T	C	C	T	C	A	G	C	T	C	C	T	A	C	C	T	T	C	T	A	C	T	T	
8	7	–	A	T	G	T	C	C	C	T	A	A	C	C	T	C	C	G	C	C	T	C	A	T	G	T	C	C	
9	6	–	A	C	G	T	C	T	C	T	G	G	T	C	T	T	C	A	C	C	C	C	A	T	A	C	T	C	
11	13	–	A	T	A	T	C	C	C	T	A	G	T	T	T	T	C	A	C	C	C	T	A	T	A	C	T	C	
12	9	–	A	T	A	T	C	C	C	T	A	G	C	C	T	C	T	A	A	T	T	T	A	C	A	C	T	T	
17	5	–	A	C	G	T	T	T	C	T	A	G	T	C	T	T	C	A	C	C	C	C	A	T	A	C	T	C	
Note:

H, haplotype; N, number of individuals per haplotype contains; R, reference sequence.

Specifically, we observed that haplotypes 3 and 11 exhibited substantial differences in body weight compared to haplotypes 5, 6, and 7 (Fig. 1A, P < 0.05). In terms of chest depth, distinctions were noted between haplotype 4 and haplotype 3, as well as haplotype 12 and haplotype 3, along with haplotype 17 (Fig. 1B, P < 0.05). Furthermore, haplotype 7 manifested variations in chest width in contrast to haplotypes 3 and 8 (Fig. 1C, P < 0.05). Notably, haplotype 17 displayed differences in tibial length when compared to haplotypes 3, 4, 5, 6, 7, 8, 9, and 12. Similarly, haplotype 7 exhibited differences in tibial length concerning haplotype 3 and 11 (Fig. 1D, P < 0.05). Additionally, haplotypes 3 and 11 displayed differences in tibial circumference when compared to haplotype 7 (Fig. 1E, P < 0.05).

Figure 1 Tested traits with different haplotypes.

Different letters indicate values are significantly different (P < 0.05).

It is worth noting that there were no significant differences observed in body oblique length among the various haplotypes. Importantly, haplotype 6 compensated for the lack of body width, resulting in no significant weight discrepancy (Fig. 1F). Further details and primary data of body measurements could be found in the Table S1.

Correlation between mtDNA copies and traits

We meticulously analyzed mtDNA copy number across the aforementioned 10 haplotypes (Fig. 2). Our analysis revealed intriguing correlation patterns between mtDNA copy number and production traits. Haplotype 17 exhibited a negative correlation with all phenotypic traits. Conversely, haplotypes 6 and 11 displayed a positive correlation with all phenotypic traits. Haplotypes 5 and 7 demonstrated positive correlations with five traits, while most traits of haplotypes 3, 4, and 8 exhibited negative correlations with mtDNA copy number. For the remaining haplotypes, correlations with production traits and mtDNA copy number were irregular.

Figure 2 Correlation between mtDNA copy number and tested traits.

** represents statistical differences between comparisons with P < 0.01. H, haplotype.

Remarkably, haplotypes 11 and 17 displayed significant positive correlations with body oblique length (P < 0.01), with haplotype 11 also showing a significant positive correlation with body weight (P < 0.01). Further details and results of mtDNA copy number could be found in the Table S2.

Maternal origin of wuliangshan black-bone chicken

In our comprehensive analysis of mitochondrial DNA sequences derived from the D-loop region across various indigenous chicken breeds, red junglefowl subspecies Gallus gallus gallus (Thailand and its neighboring areas) (Fumihito et al., 1994) and Gallus gallus spadiceus (southern Yunnan province of China) (Bao et al., 2007), we delineated six distinct haplogroups, designated HGA to HGF (Fig. 3). Haplotypes were classified in haplogroups according to the nomenclature (Achilli et al., 2012). The haplogroup HGA encapsulates a diverse array of haplotypes, including Gallus gallus gallus 1 (GGG1), GGG5, GGG4, GGG3, GGG9, GGG10, GGG11, GGG12, Gushi chicken 1 (GS1), Qingyuan chicken (QY), Xianju chicken (XJ), and Wuliangshan black-bone chicken haplotypes 6 (hap6) and 8 (hap8). Conversely, HGB comprised haplotypes such as GGG8, Gallus gallus spadiceus 2 (GGS2), GGS3, and hap7. The HGC haplogroup was characterized by the inclusion of Wuding chicken (WD), Yianjin chicken 3 (YJ3), Yianjin chicken 1 (YJ1), GS3, GS5, hap5, and hap12. The HGD group encompassed GGG2, GGG6, GGG7, Emei black chicken (EMB), Hetian chicken (HT), large bone chicken (DG), and GGS4. Haplogroup HGE was constituted by GS4, Camellia chicken 2 (CH2), GGG13, CH1, CH3, GGS5, YJ2, Baiyin ear chicken (BYE), hap4, and hap11. Lastly, HGF included GGS1, GS2, Taihe black-bone chicken (TH), Taoyuan chicken (TY), Shouguang chicken 1 (SG1) and 2 (SG2), alongside haplotypes 3, 9, and 17. The assignment of different alphabets followed by number represents various local chicken breeds and their respective haplotypes.

Figure 3 Neighbor-joining phylogenetic tree constructed from the 49 haplotypes of D-loop identified in 16 chicken populations.

Discussion

Mitochondrial DNA and its impact on economic traits

In the field of animal genetics, mitochondrial DNA (mtDNA) is a crucial tool for investigating extranuclear genetic effects on economic traits (Chen et al., 2009; Qin, Chen & Lai, 2012; Zhang et al., 2008). Point variations within the D-loop region are recognized for their potential impact on economic traits, and the polymorphism in this region provides valuable insights into the analysis of cytoplasmic genetic variation, maternal effects, and other economic traits (Oh et al., 2003). In our study, we meticulously classified 10 haplotypes based on the D-loop region, each encompassing more than five individuals, and conducted a comprehensive phenotypic analysis of Wuliangshan black-bone chickens, resulting in the categorization of these chickens into 38 haplotypes. Our observations revealed significant differences in traits such as body weight, tibial length, tibial circumference, body oblique length, chest width, and chest depth among several haplotypes. These findings align with previous research conducted on White Leghorn chickens (Li et al., 1998). Thus, it is evident that mtDNA haplotypes are associated with economic and morphological traits in chickens, as demonstrated in previous studies, including our own research (Kong et al., 2020). These genetic variations directly influence the synthesis of essential substances involved in energy metabolism, thereby impacting animal phenotypes. Since mitochondrial genome-encoded products are directly involved in oxidative phosphorylation within mitochondria, these variations are closely tied to cellular energy supply. This suggests that mitochondrial DNA variations affect energy metabolism by generating different haplotypes, ultimately leading to phenotypic variations in chickens.

mtDNA copy number and its influence on phenotypic traits

While studies on mtDNA copy number have predominantly focused on pigs and cattle, research in poultry remains scarce. Furthermore, the relationship between mtDNA copy number and different phenotypic traits in poultry has not been extensively explored. Previous studies have suggested that higher mtDNA copy number are necessary in areas with elevated ATP demand (Masuyama et al., 2005; St John, 2012), highlighting the significant impact of mtDNA copies on energy consumption. As mitochondria play a central role in supplying energy throughout an animal’s lifetime, we hypothesized that mtDNA copy number also influences phenotypic traits in chickens.

Our analysis of mtDNA copy number across ten haplotypes in poultry reveals significant insights into the genetic determinants of productive traits in egg-laying animals. Notably, the distinct correlation patterns observed between mtDNA copy number and productive traits across these haplotypes underscore the complex genetic architecture influencing phenotypic diversity. For instance, haplotype 17’s consistent negative correlation with phenotypic traits contrasts sharply with the significant positive correlations observed in haplotype 6 and 11, suggesting a nuanced role of mtDNA in regulating these traits. This is particularly evident in haplotype 11, which shows significant positive correlations with both body oblique length and body weight (P < 0.01), indicating its potential as a marker for genetic selection in breeding programs aimed at enhancing these traits. These findings align with previous research indicating the impact of mtDNA variations on the physiological and metabolic traits in chickens. For example, studies demonstrated that variations in mtDNA copy number could influence metabolic efficiency and growth rates in broiler chickens (Kim et al., 2015), suggesting a direct link between mtDNA content and energy production efficiency critical for growth and development.

Maternal origin and genetic diversity in wuliangshan black-bone chickens

As mtDNA is exclusively inherited through the cytoplasm of the egg, it retains the genetic signature of the wild ancestors over thousands of years, despite extensive crossbreeding in poultry. This genetic stability enables individuals to serve as representatives of ancestral groups, preserving the clear systematic relationships (Brown, 1981).

The inclusion of Wuliangshan black-bone chicken haplotypes within haplogroup HGA invites a focused discussion on the matrilineal origins of this unique breed. The genetic affinity of Wuliangshan black-bone chickens, as indicated by haplotypes 3, 4, 6, 8, 9, 11 and 17 to other indigenous breeds within HGA, HGC, HGE and HGF notably the several local breeds such as Gushi, Qingyuan, Xianju and etc. chickens, provides intriguing insights into their phylogenetic lineage and maternal ancestry. The clustering of Wuliangshan black-bone chicken haplotypes with those from geographically and phenotypically diverse breeds suggests a shared maternal lineage that predates the breed’s geographical and cultural isolation. This genetic evidence posits that the maternal lineage of Wuliangshan black-bone chickens may have originated from one or multiple widespread maternal ancestor common to several indigenous chicken breeds across the region. HGA, with its broad spectrum of haplotypes, suggests a significant genetic variation within this group, potentially indicative of a long-standing and widespread domestication process (Miao et al., 2013). The presence of Wuliangshan black-bone chicken haplotypes, Gallus gallus spadiceus and Gallus gallus gallus within HGB and HGE reflects their historical admixture event and the genetic connectivity and potential gene flow among different geographical populations, which is in line with the hypothesis suggested multiple origins of chickens in China (Liu et al., 2006).

Conclusions

In conclusion, our study reveals a strong association between mtDNA haplotypes and various production traits in Wuliangshan black-bone chickens. Specifically, the mtDNA copy number of haplotypes 5, 6, 7, and 11 exhibit positive correlations with these production traits, with haplotype 11 showing a significant positive correlation with body oblique length and body weight. Our findings also suggest that Wuliangshan black-bone chickens likely have multiple maternal origins, with a substantial contribution from the G. g. spadiceus subspecies. Further research is needed to unravel the intricate mechanisms underpinning the influence of mtDNA copy number on production traits in poultry.

Supplemental Information

Supplemental Information 1 Further details and primary data on body measurements, including mitochondrial DNA copy numbers, body weight, tibial length, tibial circumference, body oblique length, chest width, and chest depth, with mean values and standard errors provided for each me.

Supplemental Information 2 Raw data.

Supplemental Information 3 The sequences used in the manuscript.

Supplemental Information 4 MIQE.

Supplemental Information 5 ARRIVE 2.0 Checklist.

We thank Mr. Kai Zhang, Mr. Ran Chen (Dali Jimingjiang Breeding Poultry Co., Ltd) and Mrs. Huimin Li (Dali Animal Husbandry Workstation) for generously helping with chicken blood sample collection.

Additional Information and Declarations

Competing Interests

Author Contributions

Animal Ethics

DNA Deposition

Data Availability

The authors declare that they have no competing interests.

Wenpeng Li conceived and designed the experiments, performed the experiments, analyzed the data, prepared figures and/or tables, authored or reviewed drafts of the article, and approved the final draft.

Zhen Yang conceived and designed the experiments, performed the experiments, analyzed the data, prepared figures and/or tables, authored or reviewed drafts of the article, and approved the final draft.

Chao Yan conceived and designed the experiments, analyzed the data, prepared figures and/or tables, and approved the final draft.

Siyu Chen conceived and designed the experiments, authored or reviewed drafts of the article, and approved the final draft.

Xingbo Zhao conceived and designed the experiments, authored or reviewed drafts of the article, and approved the final draft.

The following information was supplied relating to ethical approvals (i.e., approving body and any reference numbers):

The experimental protocols and animal care were formally approved by the China Agricultural University Laboratory Animal Welfare and Animal Experimental Ethical Inspection (approval number: CAU20180619-5).

The following information was supplied regarding the deposition of DNA sequences:

The DLoop sequences are available at NCBI: PRJNA1170258

The following information was supplied regarding data availability:

The raw measurements are available in the Tables S1 and S2.

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
