# Peer review of "The relationship between mitochondrial DNA haplotype and its copy number on body weight and morphological traits of Wuliangshan black-bone chickens"

_PeerJ, doi:10.7717/peerj.17989_

## Round 0.1 · original submission · Major Revisions

Both reviewers have shown interest in this study, yet some major revisions are needed, including flaws in the experimental design, insufficient discussion, and lack of details in many sections, as the reviewers mentioned in their comments. The authors are encouraged to make substantial changes accordingly before we can move forward with a decision.

Reviewer 1 ·

Basic reporting

The authors presented a manuscript on the relationship between mitochondrial DNA haplotype and its copy number on body weight and morphological traits of Wuliangshan black-bone chickens.
The investigated topic is worthy of interest even if there are several points of concern that make the manuscript not acceptable in the present form.
General comment: several aspects, in particular about the phylogenetic analyses, have to be more in deep analysed and explained. All the manuscript needs a strong revision, trying to take in consideration the following suggestions.
More in details:
Line 5: authors should list from n. 1 to 3 for their affiliations
Lines 51-53 need a reference.
Lines 54-55, I suggest to update with more update references or using this review paper: Di Lorenzo et al. "The role of mitochondrial DNA to determine the origin of domestic chicken." World's Poultry Science Journal 71.2 (2015): 311-318.
Line 61, the sentence is arguable: the d-loop is prevalently used for phylogenetic studies based on maternal genetic diversity; anyway reference “M, 1981” is to be revised and I suggest again to include a more updated reference.
Line 64, 65 and all over the manuscript long: please carefully revise the references when reported as first letter instead of the full name “H al., 2003”; “S et al., 2012”.
Line 90: please use italics style for Gallus gallus domesticus (here and in all the manuscript long).

Experimental design

Line 101: please supply more information about Wuliangshan black-bone chickens i.e. present census, number of farms, is it endagered or under conservation plan, etc.
Line 107: it could be correct “adult body weight”?
Line 128: why the authors collected the samples just in one farm?
Line 146: the paragraph title is “PCR Amplification and DNA Sequencing”, but there are no sequencing details.
Line 158: the authors should clarify the rational of choosing VIM gene as a reference for standardization.
Line 172: authors should discuss the choice of the “Livak and Schmittgen, 2001” method for the determination of relative mtDNA copy numbers. This method was proposed for nuclear gene expression and mitochondrial copy numbers are not mentioned.
Line 176: please supply a reference for MEGA-X software. Same for the other used software in lines 177, 187 and 192.
Line 181: An NJ should be a NJ.

Validity of the findings

Results: authors should start with a paragraph dedicated to sequencing results by describing sequences lenght and range relative to the reference sequence. Then, haplotypes should be reported in a detailed table reporting mutations, haplotype ID used in the text, n. of individuals for each haplotype.
Line 221 and 225: please evaluate to replace “production” with “productive”.
Line 233: authors describe six haplogroups but they are not reported in the figures. Is the haplogroup classification standardized with the chicken mitochondrial haplogroup nomenclature? Authors should refer to universally recognized mtDNA haplogroups. Try to check: Liu et al., 2006: Y.P. Liu, G.S. Wu, Y.G. Yao, Y.W. Miao, G. Luikart, M. Baig, A. Beja-Pereira, Z.L. Ding, M.G. Palanichamy, Y.P. Zhang; Multiple maternal origins of chickens: out of the Asian jungles; Mol. Phylogenet. Evol., 38 (2006), pp. 12-19.
Lines 237-240: haplotype description should be complete and clear: for instance haplotype 4 and haplotype 11 are in the same clade in figure 3 together with GGG and GGS but they are discussed separately in the text. Please verify and include the abbreviations meaning in the figure 3.
Line 243: authors should highlight in the figure the six clusters mentioned in order to make them visible.
Line 259: “It is”, without capital letter.
Line 285: I suggest a deeper discussion on other findings such as in Ekstrand et al 2004, Gupta et al 2023.
Lines 297-299: authors state “Among these, haplotypes 5, 6, 8, 11, and 12 likely originate from the G. g. gallus subspecies, while the maternal lineage of haplotypes 3, 4, 7, 9, and 17 appears to trace back to the G. g. spadiceus subspecies”: is this highlighted in figure 3 or 4? Please, highlight relative clades in order to make clear for the readers.
Line 304-311: I suggest authors to discuss the most updated literature such as:
Miao, YW. et al 2013; Al-Jumaili et al 2020; Ceccobelli, S et al 2015; Zhang, Long et al. 2017.
Check also the manuscript Nxumalo, N., et al (2020) to improve the phylogenetic analysis section.

Additional comments

Figure 1: what is the meaning of letters above the istograms? (a, b, ab).
Table 1: G. g. spediceus should be Gallus gallus spadiceus and in Italics. Moreover, is it “(chicken)” necessary in the first column title? Finally, try to check the table template to reduce in only one (or maximum two) page/s.
Data submission: sequences should be submitted to one of data repository such as GenBank and relative accession numbers should be available in the tables.
References to be strongly checked and made compliant to the editorial guidelines.
Just as an example: C, S.J.J. (2012) or H, M., L, M.M., K, O., F, M., and S, T. (2003).

Reviewer 2 ·

Basic reporting

The manuscript descibes mitochondrial DNA haplotype and copy number variations and their possible role in production traits in chickens. Even if the manuscript is clear and well written, my overall consideration is that the text could be improved both in the description of results and discussion.
In my opinion authors should spend a couple of sentences for better describe results in term of sequencing results (such as sequences lenght and range relative to the reference sequence). Also tables should be more informative: please report haplotypes in a dedicated table reporting mutations, haplotype ID used in the text, n. of individuals for each haplotype etc.

I also encorage authors to refer to the worldwide recognized chicken mitochondrial haplogroup classification (Liu et al 2006; doi:10.1016/j.ympev.2005.09.014). I also suggest a deeper discussion based on the most updated pyhlogeny and literature both at macro- and micro-geographic level such as:
Al-Jumaili et al 2020; doi: 10.1080/10495398.2021.2000429
Ceccobelli, S et al 2015; doi 10.1016/j.livsci.2015.03.003
Gupta et al 2023; DOI: 10.1038/s41586-023-06426-5
Lasagna et al 2020, doi:10.1016/j.psj.2019.12.066;
Nxumalo, N., et al 2020; doi: 10.1080/1828051X.2020.1838350
Zhang, Long et al. 2017;. doi.org/10.1371/journal.pone.0172945

Please check References and made uniform both while citing in the text and in the reference list as described in the author’s guidelines


Figures and captions could be more detailed: for instance abbreviations should be explained, clades discussed in the text can be highlighted in the figure, and haplogroups should be reported in the phylogenetic tree.

Experimental design

As for the mtDNA copy number analyses I think that authors should justify and explain the quantification strategy used. Authors refer to Livak and Schmittgen, 2001 in which mtDNA is not mentioned. A wide, most updated, literature state that the amount of mtDNA is variable and tissue-dependent and the assessment of mitochondrial DNA copy number variation relative to nuclear DNA quantity req

Latin names have rigorous nomencalture rules, please check and follow them.

Sequences should be submitted in GenBank and relative accession numbers should be reported in the text and tables uires rigorous approach. Please discuss the rational of your chooise or test other protocols.

Validity of the findings

no comment

Additional comments

Please check the attached file

---

## Round 0.2 · Minor Revisions

Please address the remaining concerns from the reviewer.

Reviewer 1 ·

Basic reporting

First of all I want to thank the authors for their efforts in improving the manuscript that in the present revised version is more understandable for the readers.
I have just few minor comments (refferring to manuscript file with track changes):
-line 288, replace stuff with staff
-starting from line 411 and in all the manuscript long: pay attentio to latin names. The correct nomenclature is Gallus gallus (using capital letter for first letter of first name).

Experimental design

no comment

Validity of the findings

no comment

Additional comments

no comment

---

## Round 0.3 · accepted · Accept

The manuscript can now be accepted for publication.